# Unraveling the Role of Hepatic PGC1α in Breast Cancer Invasion: A New Target for Therapeutic Intervention?

**DOI:** 10.3390/cells12182311

**Published:** 2023-09-19

**Authors:** Kumar Ganesan, Cong Xu, Qingqing Liu, Yue Sui, Jianping Chen

**Affiliations:** 1School of Chinese Medicine, LKS Faculty of Medicine, The University of Hong Kong, Sassoon Road, Hong Kong, China; kumarg@hku.hk (K.G.); congx@hku.hk (C.X.); lqq2022@hku.hk (Q.L.); u3006765@connect.hku.hk (Y.S.); 2Shenzhen Institute of Research and Innovation, The University of Hong Kong, Shenzhen 518057, China

**Keywords:** PGC1α, lipogenesis, NAFLD, breast cancer, cell invasion, ERRα

## Abstract

Breast cancer (BC) is the most common cancer among women worldwide and the main cause of cancer deaths in women. Metabolic components are key risk factors for the development of non-alcoholic fatty liver disease (NAFLD), which may promote BC. Studies have reported that increasing PGC1α levels increases mitochondrial biogenesis, thereby increasing cell proliferation and metastasis. Moreover, the PGC1α/ERRα axis is a crucial regulator of cellular metabolism in various tissues, including BC. However, it remains unclear whether NAFLD is closely associated with the risk of BC. Therefore, the present study aimed to determine whether hepatic PGC1α promotes BC cell invasion via ERRα. Various assays, including ELISA, western blotting, and immunoprecipitation, have been employed to explore these mechanisms. According to the KM plot and TCGA data, elevated PGC1α expression was highly associated with a shorter overall survival time in patients with BC. High concentrations of palmitic acid (PA) promoted PGC1α expression, lipogenesis, and inflammatory processes in hepatocytes. Conditioned medium obtained from PA-treated hepatocytes significantly increased BC cell proliferation. Similarly, recombinant PGC1α in E0771 and MCF7 cells promoted cell proliferation, migration, and invasion in vitro. However, silencing PGC1α in both BC cell lines resulted in a decrease in this trend. As determined by immunoprecipitation assay, PCG1a interacted with ERRα, thereby facilitating the proliferation of BC cells. This outcome recognizes the importance of further investigations in exploring the full potential of hepatic PGC1α as a prognostic marker for BC development.

## 1. Introduction

Non-alcoholic fatty liver disease (NAFLD) is the leading cause of liver disease in high-income countries, affecting 25–40% of the population [1]. The prevalence of NAFLD has been increasing worldwide and has become a key issue for personal and public well-being [2]. It is characterized by widespread clinicopathological settings, including intracellular fat deposition (steatosis) and nonalcoholic steatohepatitis (characterized by severe steatosis, fibrosis, and inflammatory necrosis). It occurs as a result of liver accumulation of excessive triglycerides, cholesterol, and fatty acids, and dysregulation of cholesterol homeostasis [3,4]. Recent large meta-analyses have reported that NAFLD is associated with an increased long-term risk of developing chronic kidney disease (~1.45-fold) [5], cardiovascular disease [6], diabetes (~2.2-fold) [7], and extrahepatic cancers (esophageal, stomach, pancreas, colon, lung, breast, and gynecological cancers) (~1.2-fold to 1.5-fold) [8]. Remarkably, studies found that NAFLD is possibly related to a higher incidence of breast cancer (BC) [9,10]. However, it remains uncertain whether NAFLD is considerably connected to BC risk. A recent nationwide population-based study revealed that postmenopausal women with a higher fatty liver index showed a higher incidence of BC [11]. Several studies reported a direct association between metabolic syndrome and BC risk in postmenopausal women [12,13,14]. Despite having a normal BMI, postmenopausal women who are metabolically unbalanced or have central adiposity may be more susceptible to BC [12].

BC is the most common cancer affecting women worldwide and the leading cause of cancer-related deaths among women [15]. Known risk factors for BC include nulliparity, late pregnancy, early menarche, late menopause, and hormonal or reproductive factors [16]. The incidence of BC is now rapidly rising in Asia and other continents of the world. The westernization of lifestyles with changes in reproductive behavior is the primary justification for this rise [17]. Metabolic syndrome and its components have also been found to be linked to an increased risk of BC, according to recent meta-analyses [18]. Obesity has also recently been recognized as a BC risk factor [19]. Metabolic abnormalities and obesity are shared common risk factors for both NAFLD and BC [20]. Clinical studies have shown that BC patients with NAFLD have poorer prognoses according to recurrence. Nearly 63% of newly diagnosed patients with BC have NAFLD [9] and there is a significantly higher incidence of NAFLD in patients with BC than in healthy controls (45.2% vs. 16.4%) [21]. However, the exact incidence of NAFLD in women with BC remains uncertain, and the effect of NAFLD on BC prognosis has yet to be recognized. Based on the shared connotations of these metabolic components, it is rational to hypothesize an association between NAFLD and BC. Furthermore, there are no preclinical studies available to connect directly both NAFLD and BC.

Peroxisome proliferator-activated receptor gamma coactivator 1 alpha (PGC1α) is a transcription factor coactivator primarily expressed in the liver, adipose tissue, and skeletal muscle, and influences a majority of cellular metabolic pathways [22]. Their versatile actions are achieved by tissue-specific interactions with different transcription factors and nuclear receptors. Under normal conditions, hepatic PGC1α expression is relatively low but is highly inducible in hepatotoxic conditions, diabetic, and ob/ob mouse models [23]. Earlier, it was demonstrated that sequential cascades of TXNIP-PRMT1-PGC-1α are involved in the development of hepatic steatosis [24], and co-activate with sterol-responsive element-binding proteins to stimulate lipogenic gene expression [25].

In addition to its effects on various metabolisms, PGC1α is involved in promoting angiogenesis in BC [26]. It promotes ERBB2+ tumor growth in vivo by regulating mitochondrial metabolism and angiogenic activities [27,28]. Its effects on these processes contribute to the tumor’s metabolic adaptations and ability to sustain its growth in the surrounding tissue. It allows neovascularization in mammary tumors, thereby increasing the availability of nutrients and resulting in tumor growth [29]. Studies have reported that increasing PGC1α levels increase mitochondrial biogenesis, thereby increasing metastasis. Patients with bone metastases have higher PGC1α levels in their mammary tumors, and there is a negative correlation between PGC1α expression and patient survival [26,30]. Similarly, a study demonstrated that PGC1α is positively associated with the glutamine pathway in ERBB2+ BC patients, and elevated expression of this pathway is related to decreased patient survival [30]. In addition, the PGC1α/ERRα (estrogen-related receptor alpha) axis is an important regulator of cellular metabolism and genes involved in mitochondrial biogenesis in a variety of tissues, including BC. PGC-1a and ERRα levels are highest in triple-negative and HER2+ BC, which have the worst prognosis [26,30]. However, no studies on the luminal BC-related PGC-1a/ERRα axis have been conducted. The exact prevalence of NAFLD in women with BC is still unknown, and how it affects BC prognosis is not yet known. It makes sense to speculate that NAFLD and BC are related given the shared associations of these metabolic components. In addition, preclinical research has not directly linked NAFLD to BC. Therefore, the present study aimed to determine whether hepatic PGC1α promotes BC cell invasion through ERRα.

## 2. Materials and Methods

### 2.1. Chemicals

Palmitic acid (PA), 3-[4,5-dimethylthiazole-2-yl]-2,5-diphenyltetrazolium bromide (MTT), ERRα antagonist, XCT-790, Recombinant PGC1α, and common chemicals were purchased from Sigma Aldrich (St. Louis, MO, USA). All chemicals were obtained from Sigma-Aldrich except DMEM, Insulin-Transferrin-Selenium, and FBS (Life Technologies, Carlsbad, CA, USA).

### 2.2. Cell Culture

Normal hepatocytes, AML12 (alpha mouse liver 12), and MIHA (Immortalized human hepatocytes), human breast cancer cell lines (MCF-7 cells), and mouse mammary carcinoma cell lines (T47D, E0771, and 4T1) were purchased from American Type Culture Collection (ATCC, Manassas, VA, USA; passage number: 5–15) and maintained according to their instructions. The MIHA and E0771 cells were cultured in Dulbecco’s modified Eagle’s medium (DMEM, 1 g/L glucose) and Ham’s F12 1:1 medium with 0.005 mg/mL insulin, 0.005 mg/mL transferrin, 5 ng/mL selenium, and 40 ng/mL dexamethasone (Gibco, Grand Island, NY, USA). AML12 cells were grown in DMEM-F12; MCF-7 cells were cultivated in Eagle’s Minimum Essential Medium (Gibco, Grand Island, NY, USA); 4T1 and T47D cells were cultured in Roswell Park Memorial Institute (RPMI)-1640 medium (Gibco, Grand Island, NY, USA); and all cells were seeded to reach about 80% confluence prior to experimental treatment. All these cells were supplemented with 10% fetal bovine serum (Gibco, Grand Island, NY, USA), streptomycin (100 μg/mL), and penicillin-streptomycin (FBS, 100 U/mL, Sigma-Aldrich, St. Louis, MO, USA) at 37 °C in a humidified atmosphere of 95% air, 5% CO_2_.

### 2.3. Fatty Acid Treatment

Palmitic acid (PA) powder was dissolved in isopropanol and complexed with fatty acid-free bovine serum albumin (10% *w*/*v*) that was dissolved by shaking gently overnight at 37 °C to yield an 8 mM solution. To induce lipid accumulation in hepatocytes, AML12 and MIHA cells were treated with PA at various concentrations (0.2, 0.4, and 0.8 mM) for 12, 24, and 48 h. DMEM F12 containing BSA was used as a control. For lipogenic marker analysis, cells were harvested with trypsin-EDTA (Gibco, Grand Island, NY, USA), and prepared proteins were subjected to western blotting.

### 2.4. Cell Viability Assay of Hepatocytes and BC Cells

Hepatocytes were plated in 96-well plates at a density of 3–5 × 10^3^ /well and cultured for 12 h. Then, the cells were treated with different concentrations of PA (0.2, 0.4, and 0.8 mM) for 12, 24, and 48 h and 50 μL of MTT was added into the media for 4 h. Similarly, BC cells were treated with different concentrations of recombinant PGC1α (0, 5, 10, 20, and 40 ng/mL). The media were removed and DMSO was added to each well. The absorbance was measured in a microplate reader (BioTek, Winooski, VT, USA) at 570 nm, and the relative cell viability of hepatocytes was recorded as a ratio between the normal and BSA control, and the PA treatment groups were compared only to the BSA control. Three replicate wells were counted for each condition.

### 2.5. Oil Red O Staining

Hepatocyte cells were fixed in 4% paraformaldehyde in PBS for 10 min and washed with 60% isopropanol. Then, the cells were stained with 0.2% Oil Red O (ORO) in 60% isopropanol for 10 min. After washing with water, hematoxylin stains were added for 30 sec and washed again with water. Cells were imaged using a Leica DM IRB microscope (Leica DM IRB; Leica Microsystems Inc., Deerfield, IL, USA). After staining with ORO and hematoxylin, it was washed three times with DD water. Isopropanol (100%) was added to the extract ORO stain for five minutes, while gently rocking on the surface. Read the absorbance at 492 nm using a microplate reader.

### 2.6. Transfection

After stabilizing AML12 and MIHA cells for 24 h, plasmid DNAs was transfected with GeneExpresso Max transfection reagent (Excellgen, Rockville, MD, USA) following the manufacturer’s instructions. The primers used were indicated in Appendix A. After 24 h, the culture medium was replaced with a serum-free medium and treated with drugs as indicated for 24 h. For siRNA transfection, PGC1α or scrambled siRNA was transfected using Lipofectamine RNAiMAX (Invitrogen, Carlsbad, CA, USA) following the manufacturer’s instructions.

### 2.7. Western Blotting

Western blot assay was conducted as previously described [17]. The proteins from the cell were lysed in RIPA buffer (pH = 7.4) comprised of a protease inhibitors cocktail (10 μg/mL, Cat# 5872S, Cell Signaling Technology, Danvers, MA, USA). The contents were centrifuged at 12,000× *g* at 4 °C for 20 min, and the concentration of protein in the supernatants was determined using Bradford reagent (BioRad, Hercules, CA, USA) with bovine serum albumin (BSA, Sigma Aldrich, St. Louis, MO, USA) as the standard. The protein samples were separated by electrophoresis on SDS-PAGE 10% or 12.5% gels. After being blocked in 3% BSA, the membrane was incubated with primary antibodies (Appendix A). For secondary antibodies, anti-mouse and anti-rabbit were used. To visualize protein bands, a chemiluminescence (ECL) system (Cat# WBLUF0500, Millipore, Burlington, MA, USA) was used.

### 2.8. Immunoprecipitation (IP) Assay

An Immunoprecipitation Kit (Thermo Scientific, Waltham, MA, USA, catalog #100007D, USA) was used for the IP assay according to the manufacturer’s instructions. Briefly, lysis buffer and protease inhibitor cocktail (Thermo Scientific, USA) were used to lyse the rPGC1α-treated BC cells. The protein supernatant was extracted by centrifugation at 4 °C and 13,000 rpm for 30 min after incubation on ice for more than 30 min. For pre-clearing, protein-G agarose beads were added and rotated for three hours at 4 °C. We set 100 μL as the input group. Four microliters of control immunoglobulin G (Thermo Scientific, USA) or 8 μL of anti-PGC1α (1:50, Thermo Scientific, USA) was added to the cell lysate (400 μL per tube) at 4 °C for one hour while shaking. Then, 50 µL of protein-G-agarose beads (Thermo Scientific, USA) was added to each tube and the mixture was rotated overnight at 4 °C. The following day, immunoprecipitates were washed with the wash buffer which had already been provided in the kit and resuspended in 20–50 μL of 2× western loading buffer. Standard Western blotting methods were used to determine the expression of the proteins of interest (ERRα) in the input, IgG, and PGC1α-immunoprecipitated samples.

### 2.9. Detection of PGC1α by ELISA

ELISA is an efficient and effective method to assess the expression level of PGC1α in PA-treated conditioned media samples. The concentrations of PGC1α in CM were determined using ELISA kits according to the manufacturer’s instructions (Bovine peroxisome proliferator-activated receptor gamma coactivator 1α ELISA kit, Cat. No# RK07760, Abclonal, Woburn, MA, USA). The sample dilution used for media was analyzed as a blank control. The optical density (OD) of each well was measured at a wavelength of 450 nm in a microplate reader (Thermo Scientific, Waltham, MA, USA). The concentration of PGC1α was calibrated with the PGC1α standard curve. Assays were repeated in triplicate.

### 2.10. Colony Formation Assay

Approximately 1000 BC cells were seeded in a six-well plate and treated with different concentrations of recombinant PGC1α (0, 5, and 40 ng/mL) and then provided with fresh medium for at least a week. The cells were stained with crystal violet (Sigma-Aldrich) for 10 min after being fixed with methanol for 15 min at room temperature. As previously stated [31], quantification of formed colonies was performed. The clonogenicity of cells treated with recombinant PGC1α was determined by comparing the number of colonies formed in the treatment groups to those in the control group (represented as the percentage of control).

### 2.11. Wound Healing Assay

The ability of a cell to migrate is demonstrated by the wound healing assay. Cells were spread evenly in a 12-well plate, and cell density was maintained at 90–100 percent coverage after short-term culture. Cut the cell layer gently with sterile tips. PBS was used to clean the cells. A new medium containing 1% FBS was used to replace it, and the cells were photographed under the microscope at 0 and 24 h. The scratches were determined using Image J 1.53 version (Image J, NIH, Bethesda, MD, USA) and standardized to the typical scratch region at 0 h.

### 2.12. Transwell Migration Assay

Cell migration abilities were determined in different concentrations of recombinant PGC1α (0, and 40 ng/mL) using a coated transwell assay with Matrigel (BD Biosciences, Franklin Lakes, NJ, USA). A total of 1 × 10^5^ BC cells suspended in 200 μL serum-free medium were seeded into the upper transwell chambers, and the complete medium was added to the lower compartment of the transwell chamber (Corning, #3422, Corning, NY, USA) in a 24-well plate. The plate was then placed back into the incubator for 48 h. After carefully wiping the non-migrating cells inside the transwell chamber, the cells on the outside were fixed with 4% PFA and stained with 0.2% crystal violet. After cleaning the transwell chamber three times with PBS, images were taken under a microscope (Leica, Wetzlar, Germany).

### 2.13. Analysis of Biological Information

The association between PGC1α and BC, as well as overall survival was performed by the online tool KM plot (http://kmplot.com/ (accessed on 11 May 2023)) [32] with the Affymetrix ID: PPARGC1 (219195_at). Differential gene expression analyses of the tumor, normal, and metastatic tissues were conducted by the online tool TNMplot (https://www.tnmplot.com/ (accessed on 11 May 2023)) with genes’ symbols based on RNA-Seq data offered by the database [33].

### 2.14. Statistical Analysis

The data were presented as means ± SD for three replicates. Data were analyzed with one-way ANOVA followed by group comparisons using a post hoc Tukey’s multiple comparison test using GraphPad Prism 7 (GraphPad Software, San Diego, CA, USA).

## 3. Results

### 3.1. Relationship between PGC1α Expression in Postmenopausal Women and Overall Survival Curves

Currently, there is much interest in confirming prognostic or projecting candidate genes in highly effective BC cohorts. Based on the online Kaplan–Meier plotter tool, we drew survival plots, which were used to assess the relevant expression levels of PGC1α on the clinical outcome of BC individuals. Using the selected parameters, analysis was performed on PPARGC1 (219195_at), which represented 1879 patients. The hazard ratio of PGC1α was 1.29 (1.04–1.61, *p* = 0.023), with the median overall survival (OS) of the respective low and elevated expression cohorts of 20 and 60 months (Figure 1A). Patients with BC and high levels of PGC1α expression had poor survival rates. The Palliative Performance Scale (PPS) was used to estimate survival and assess patients’ symptoms. Our results reaffirmed PPS as a significant predictor of survival, with decreased survival times associated with higher PPS levels (Figure 1B). Kaplan–Meier curves comparing estimated survival probabilities between the low and elevated PGC1α groups were stratified by tumor subtype and menopausal status (Figure 1C). The results indicated that elevated PGC1α expression occurred in the postmenopausal group (*n* = 150, *p* = 0.022). Using the Cancer Genome Atlas (TCGA) database, we examined the expression and clinical significance of PGC1α expression in the primary tissues of BC patients, and the results showed that patients with BC had higher PGC1α expression than normal controls (Figure 1D).

### 3.2. PA Treatment Decreases the Viability of Hepatocytes

In the present study, our main focus is on the effects of palmitic acid (PA) on hepatocytes. PA is a well-known saturated fatty acid that is associated with an increased risk of metabolic diseases, including non-alcoholic fatty liver disease (NAFLD). It has been demonstrated to induce lipotoxicity in hepatocytes, leading to inflammation, oxidative stress, and non-alcoholic steatohepatitis (NASH). As such, PA causing NAFLD and NASH serves as an appropriate model for our study. Furthermore, several in vitro and in vivo studies have implicated the role of PA in promoting hepatotoxicity characterized by severe steatosis, fibrosis, and inflammatory necrosis [34,35,36,37]. To determine the cytotoxicity of PA on normal hepatic cell lines, we performed an MTT assay on AML12 and MIHA cell lines for 12, 24, and 48 h, respectively (Figure 1E,F). Treatment of hepatocytes with different concentrations of PA (0.2, 0.4, and 0.8 mM) for variable time intervals significantly decreased the cell viability in both hepatic cell lines compared to the BSA and normal control.

### 3.3. PA Treatment Promotes Lipid Accumulation in the Liver Cells

Toward establishing an in vitro model of hepatic steatosis, the effect of varying concentrations of PA (0.2, 0.4, and 0.8 mM) was studied on the extent of lipid accumulation in hepatocytes for 12, 24, and 48 h, respectively. The AML12 and MIHA cells were initially grown in a T-75 flask and later plated at a density of 2 × 10^4^ cells per well in a 96-well plate. After PA treatment, intracellular lipid droplets in the hepatocytes were stained red, and nuclei were stained blue, which was visualized under a microscope (Figure 1G). Lipid droplets were greater in number in the dose-dependent manners of PA treatment (high magnification) when compared to normal and BSA control. ORO is being used for semi-quantitative analysis of lipids in the cell lines [38,39,40]. To enable quantitative measurements, the ORO dye was eluted from the cells using isopropanol, and absorbance was photometrically determined at 492 nm. The lipid content was significantly higher in a dose-dependent manner of a PA treatment when compared to normal and BSA control (Figure 1H).

### 3.4. PA Treatment Promotes PGC1α Expression and Lipogenic Markers in the Hepatocytes

Since PGC1α is a transcriptional co-activator that modulates diverse aspects of metabolism in the liver, which may be stimulated by treatment with high fat, we speculated whether PA treatment mediates the expression of PGC1α. To determine the expression of PGC1α in PA-treated hepatocytes, we performed western blot analysis (Figure 2A). The dose-dependent manner of PA (0.2, 0.4, and 0.8 mM) treatment substantially increased PGC1α expression compared to BSA and normal controls (Figure 2A,B). PA treatment of hepatocytes typically induced lipogenic proteins such as acetyl-CoA carboxylase (ACC), fatty acid synthase (FAS), stearoyl CoA desaturase-1 (SCD1), lipoprotein lipase (LPL), and liver-type fatty acid-binding protein (FABP-L) (Figure 2C,E). In addition, PA increased the transcriptional co-activator PGC-1α and inflammatory markers, such as nuclear factor-kB (NF-kB), cyclooxygenase-2 (COX-2), tumor necrosis factor-alpha (TNF-α), and interleukin-6 (IL-6) (Figure 3A,C). The intensity of lipogenic and inflammatory marker expression was significantly higher after 24 h treatment with PA compared to the respective controls (Figure 2D,F and Figure 3B,D). These results suggest a positive relationship between PA-induced PGC-1α expression and lipogenic proteins, as well as NF-kB activation.

### 3.5. PA-Treated Hepatic Cell-Derived Conditioned Medium Promotes BC Growth

Interestingly, in this study, PA-treated hepatic cell-derived conditioned medium (CM) significantly increased BC cell growth. For this purpose, we quantified the levels of PGC1α in the conditioned medium (CM) using ELISA. CM was prepared from hepatic cells that were previously treated with 0.8 mM PA for 6, 24, and 48 h, followed by a 12 h treatment with a fresh complete medium. The CM was filtered and diluted with 25% fresh medium for further analysis. There were significantly elevated levels of PGC1α in the CM at all time points compared to those in the control. The levels of PGC1α in the CM were elevated at 48 h, according to ELISA results (Figure 4E). To determine the proliferative effect of CM (obtained from PA-treated AML 12 and MIHA), we performed an MTT assay on MCF7, E0771, and 4T1 BC cell lines for 24 h (Figure 4A,B). AML12 (Figure 4A) and MIHA (Figure 4B)-derived CM significantly increased BC cell growth compared with the normal control.

### 3.6. Recombinant PGC1α Promotes BC Cell Proliferation and Decreases Apoptosis-Related Factors

Using recombinant PGC1α (rPGC1α), we analyzed the effectiveness and accuracy of PGC1α levels in BC cell proliferation. Treatment with rPGC1α (0, 5, 10, 20, and 40 ng/mL) significantly increased the cell growth of BC cells, E0771 (Figure 4C), and MCF7 cells compared to the control (Figure 4D), which was analyzed by MTT assay. A clonogenic assay is an in vitro cell survival assay based on the capacity of a single cell to develop into a colony. This assay is also known as the “colony formation assay”. The effectiveness of cytotoxic agents and cell proliferation can be assessed using this test. BC cells treated with rPGC1α (0, 5, and 40 ng/mL) produced a significantly higher number of colonies than those in the control group (Figure 4G,H). In addition, treatment with rPGC1α (40 ng/mL) significantly reduced the levels of apoptosis markers (BAX, cytochrome c, caspase-9, cleaved caspase-9, and caspase-3) and increased Bcl-2 and PARP1 expression in both the BC cell lines (Figure 5A,B). In addition, rPGC1α (0, 5, and 40 ng/mL) significantly increased the proliferation of downstream pathways (PI3K-mTOR and Ras-ERK1/2) compared to the control group (Figure 5C,D).

### 3.7. Silencing of PGC1α Inhibits BC Cell Proliferation and Increases Apoptosis-Related Factors

Figure 5E shows two sets of PGC1α siRNA transfection and the absence of PGC1α expression when compared to a blank and negative control (scrambled). Silencing of PGC1a prevents the growth of all studied luminal BC cells (Figure 5F). SiPGC1α decreased the expression of proliferation markers Ki67 and PCNA (Figure 6A,B) compared to the scrambled. Similarly, PGC1α silencing significantly increased the levels of apoptosis markers (BAX, cytochrome c, caspase-9, cleaved caspase-9, and caspase-3) and decreased Bcl-2 and PARP1 expression in both BC cell lines (Figure 6C,D). In addition, siPGC1α significantly decreased the number of colonies compared with that in the negative control group (Figure 6E,F).

### 3.8. Recombinant PGC1α Promotes Expression of ERRα

Together with the estrogen-related receptor (ERRα), PGC1α is a crucial regulator of cellular metabolism and energy homeostasis that regulates mitochondrial and metabolic gene networks. ERRα dimerization is necessary for the interactions between PGC-1 and other coactivators, which ultimately results in transactivation. Initially, we determined whether PGC1α treatment with dose-dependent manners of rPGC1α in BC cells. For this purpose, we examined ERRα expression levels after treatment with rPGC1α (0, 5, 40 ng/mL) using western blot. The results showed that rPGC1α treatment significantly expressed ERRα in both BC cell lines (Figure 7A). Treatment of rPGC1α (0, 5, and 40 ng/mL) significantly increased the expression of ERRα in both BC cell lines, in a dose-dependent manner (Figure 7B). Furthermore, it was also proved that PGC1α interacted with ERRα in the co-IP assay (Figure 4F). This study indicated that ERRα binds to PGC1α and is involved in various cellular processes.

### 3.9. ERRα Antagonist Inhibits PGC1α-Induced BC Proliferation

High expression of PGC1α in BC is closely connected with plenty of biological processes, including proliferation, angiogenesis, cell motility, and metastasis [28,30,41]. XCT-790, an ERRα antagonist, was used in this study to determine whether ERRα coactivates PGC1α and modulates BC proliferation. Treatment with dose-dependent manners of rPGC1α (0, 5, and 40 ng/mL) and XCT-790 (5 μM) decreased the proliferation of downstream pathways (PI3K-mTOR and Ras-ERK1/2) compared to that in the control group (Figure 7C,D). Thus, this study indicates that ERRα coactivates PGC1α and promotes BC cell proliferation.

### 3.10. Recombinant PGC1α Promotes Migration, Invasion, and EMT Process

As previously demonstrated [26,30], PGC1α is closely related to the invasion and migration ability of BC cancer cells. Here, the effect of PGC1α on cell motility was also examined by in vitro methods of wound healing assay, Matrigel-uncoated transwell cell migration, and Matrigel-coated transwell cell invasion assays. Treatment with rPGC1 (40 ng/mL) significantly increased wound healing rate (Figure 8A,B), cell migration (Figure 8C), and invasion (Figure 8D) abilities in both BC cell lines compared to control. EMT (epithelial–mesenchymal transition) is a biological process by which epithelial cells lose their cell–cell adhesion and cell polarity and undergo changes in cell-matrix adhesion. As a result of EMT, invasive and migration characteristics are enhanced. Treatment with rPGC1 (40 ng/mL) significantly increased EMT markers compared to control (Figure 8E,F). These results indicate that PGC1α directly promotes invasion and migration as well as EMT process in BC cells.

### 3.11. Silencing of PGC1α Inhibits Migration, Invasion, and EMT Process

On the other hand, siPGC1α inhibits invasion, migration, and EMT processes in both BC cells. Silencing of PGC1α significantly decreased wound healing rate (Figure 9A,B), cell migration (Figure 9C), and invasion (Figure 9D) abilities in both BC cell lines compared to control. Similarly, siPGC1 significantly decreased EMT markers compared to control (Figure 9E,F). These results indicate that PGC1α enhances invasive and migratory abilities in both BC cells.

## 4. Discussion

It is generally accepted that dividing cells, including BC cancer cells, use cellular metabolism to meet their energy requirements. Adaptive metabolic reprogramming, which is triggered in part by oncogenic transformation, aids the proliferation of tumor cells during growth [42]. Rapidly proliferating BC cells undergo autonomous metabolic reprogramming, which encourages self-sustaining signal transduction mechanisms to support their growth and survival [43]. The main causes of BC cancer patient deaths and metastases may be hindered by therapeutic strategies that are informed by a better understanding of the energy requirements of invading cancer cells [44]. To identify mechanisms of BC invasion, we conducted preclinical experiments to investigate whether a metabolic key regulating factor that involves distinct energy needs for cell proliferation and invasiveness of the cancer cells. Studies have reported that obesity, diabetes, hypercholesterolemia, and NAFLD have great impacts on the risk of BC and the prognosis of BC patients in several ways [45,46,47]. Earlier, a retrospective cohort study also demonstrated that the incidence of NAFLD in patients with BC is considerably higher than in healthy individuals [46]. NAFLD is independently associated with the development of BC risk [11,21]. Moreover, BC patients with NAFLD revealed poorer prognosis in the context of recurrence [48,49]. Hence, diagnostic assessment for NALFD and its metabolic factor is significant in dealing with BC.

NAFLD is an advanced, progressive metabolic disease that arises globally. Hence, the requirement for cellular models of human steatosis and high-throughput assays for determining intracellular lipid levels is ever-increasing. A high concentration of PA represents a cellular model of steatosis which promotes acute toxicity due to over accumulation of fat in the liver [50,51]. Lipotoxicity is implicated in the pathogenesis of non-alcoholic steatohepatitis and saturated long-chain fatty acids are the major contributor to lipotoxicity [52]. PA treatment causes cellular toxicity and low viability resulting in the pathogenesis of hepatic steatosis [53]. Moreover, this toxicity was time and doses of exogenously added fatty acid-dependent [54]. The outcome of the present study is also consistent with the earlier reports that PA treatment provides acute liver toxicity, which is time (12, 24, and 48 h) and dose-dependent (0.2, 0.4, and 0.8 mM). PA was initially complexed with BSA prior to the addition to media, which prevented the fatty acid aggregation and considerably improved the solubility [55]. As a result, PA was taken up more efficiently by the hepatocytes and fattened within 12 h and increased the toxicity when increasing the incubation time [53]. The time-course fatting of hepatocytes indicated that cell viability decreased after incubating cells with 0.8 mM PA for 24 and 48 h, however, cytotoxicity was detected as quick as 12 h of incubation.

Accumulation of intracellular lipid levels is often visualized and quantified using ORO staining, which is a simple and reliable technique [40,56,57,58]. In the present study, we observed more lipid droplet accumulation at high concentrations of PA-treated hepatocytes. PGC1α is an upstream molecule in the progression of hepatic steatosis. In the present study, we found that PA treatment elevated PGC1α levels in the hepatocytes and was involved in lipogenic signaling pathways and inflammation in vitro. PGC1α is a critical metabolic regulator in various tissues, particularly involved in carbohydrate and lipid metabolism in the liver [24,52]. In this study, PGC1α was significantly increased in the dose-dependent manners of PA-treated hepatocytes and is implicated in the pathophysiology of hepatic lipogenesis and inflammation.

Previously, co-culture models have been developed, which are more accurate, mimic structural and functional changes in breast architecture, and increase their use as preclinical screens of therapies [59]. In the present study, we used CM obtained from PA-treated hepatocytes, which were treated with BC cell lines, resulting in robust cell proliferation. Although studies have supported a link between NAFLD and BC [10,60], our in vitro study also supports this link by examining the link between PA-induced fatty liver and BC via PGC1α. However, to validate this hypothesis, animal studies are necessary. Several clinical studies have linked metabolic syndrome to BC risk in postmenopausal women [12,61,62,63]. Figure 1C also shows that PGC1α expression is very high in postmenopausal women; it is also closely associated with BC risk and reduced overall survival. More recently, NAFLD has been identified as a possible contributor to BC progression in women [60].

This study found that PGC1α mediated BC cell proliferation, migration, and decreased apoptosis-related factors are functionally relevant for invasive dissemination. Invading cancer cells relied on PGC1α to accelerate the EMT during transit. Silencing of PGC1α significantly impaired BC proliferation, invasion, and migration, decreased the frequency of EMT processes, and promoted apoptosis-related factors. Our results are consistent with an earlier investigation that found that invasive BC showed a robust connection between PGC1α and the growth of distant migration [42]. Although it has to be highlighted that more animals and clinical investigations are needed to authenticate whether PGC1α induces invasion and metastasis, these in vitro outcomes make evident that the invasion and migratory properties of BC cancer cells are dependent on PGC1α and recognize PGC1α as a possible target for therapeutic intervention.

A transcriptional regulatory node crucial for the expression of genes involved in metabolism is formed by the interaction of PGC1α and ERRα [64]. As a key regulator of energy metabolism, the PGC1α/ERRα axis is now well established. PGC1α/ERRα axis is a central regulator of metabolism in cancers, notably BC [28]. ERRα and PGC1α have highly adaptable expression and activity that can react to a variety of physiological and pathological cues [65]. This study provides further evidence that PGC1α is a crucial component of the energy-sensing signaling cascade [64]. Estrogen can exert its action through an extranuclear estrogen receptor pathway that is implicated in cell proliferation, migration, and apoptosis, particularly in BC; however, its function and mechanisms are not fully understood [66]. Our findings indicate that ERRα binds to PGC1α, promoting BC proliferation through the downstream expression of RAS-RAF-MAPK and PI3K-Akt-mTOR pathways. Treatment with the ERRα antagonist (XCT-790) prevents this binding, thereby inhibiting the protein markers associated with BC proliferation. Earlier clinical analyses of human invasive BC revealed a significant correlation between PGC1α expression in invasive cancer cells and the formation of distant metastases. Notably, the expression of the ERRα-PGC1α-axis in invasive ductal carcinoma patients is also correlated with poor prognosis [67]. Our in vitro results support previous findings that the ERRα-PGC1α axis promotes BC cell proliferation.

The present study suggests that PGC1α has both extrinsic and intrinsic effects that promote BC. The hepatic PGC1α circulates and binds with ERRα, which is known to regulate metabolism in BC and is considered an extrinsic effect. Our findings indicate that ERRα binds to PGC1α, promoting BC proliferation through the downstream expression of RAS-RAF-MAPK and PI3K-Akt-mTOR pathways. In addition, we examined the effects of intrinsic PGC1α expressed in BC cells through siRNA-mediated knockdown experiments. Our findings are consistent with earlier studies that suggest PGC1α derived from tumor intrinsic sources also plays a role in BC [42,68,69,70]. Based on these findings, we explicitly state that both tumor intrinsic and extrinsic mechanisms may contribute to the malignant transformation of BC cells.

The crucial question is whether hepatic PGC1α is reliable enough to be used as a marker to predict prognosis in patients with BC. We expected to concentrate on hepatic PGC1α because of the following factors. First, it has been hypothesized that hepatic PGC1α expression is linked to the invasiveness of luminal BC cells and tumorigenesis in the mammary gland; however, the function of PGC1α as a biomarker in a non-selected BC population has not yet been established, according to the earlier studies. Second, considering that PGC1α is involved in several cellular pathways, we investigated whether hepatic PGC1α exhibited prognostic characteristics in BC. Based on our in vitro results, we found that hepatic PGC1α contributes to the growth, invasion, migration, and tumorigenesis of BC. Animal and human studies are still required to validate our in vitro findings.

## 5. Conclusions

The functions of PGC1α involve its role as a tumor-promoting factor. PA stimulates the expression of PGC1α as well as lipogenesis and inflammation. The CM obtained from PA-treated hepatocytes significantly increased BC cell proliferation. Similarly, treatment with rPGC1α in the BC cells increased cell growth, migration, and invasive behavior. The silencing of PGC1α showed a decreasing trend. PGC1α interacts with ERRα, thereby facilitating BC cell proliferation. Targeting the metabolic PGC1α-ERRα axis may be a potentially effective candidate for BC treatment.

## Figures and Tables

**Figure 1 cells-12-02311-f001:**
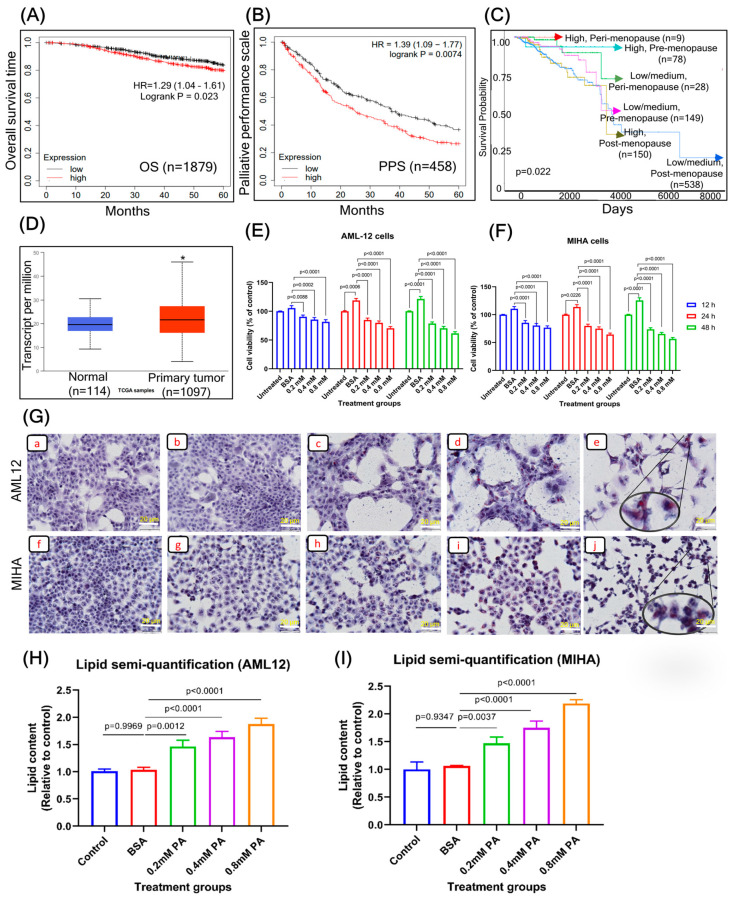
(**A**) Overall survival of patients with BC correlates to the expression of PGC1α. (**B**) The survival curves from the KM plotter show that lower PPS levels are associated with reduced survival. (**C**) Cancer Genome Atlas (TCGA) database curves of menopausal status in BC patients with survival. According to the results, the postmenopausal group (n = 150, *p* = 0.022) exhibited increased expression of PGC1α. (**D**) clinical significance of PGC1α expression in patients with BC. * *p* < 0.05. (**E**,**F**) PA (0.2, 0.4, and 0.8 mM) treatment promotes liver toxicity, which was tested for 12, 24, and 48 h using cell viability assay. The data were presented as means ± SD for three replicates. Data were analyzed with a one-way ANOVA followed by group comparisons using a post hoc Tukey’s multiple comparison test. (**G**) Exposure to PA results in intracellular lipid accumulation in hepatocytes. AML (**a**–**e**), and MIHA (**f**–**j**) were treated with PA (0.2, 04, 0.8 mM) for 24 h and were stained with Oil Red O, and nuclei were stained with hematoxylin, under microscopic observation at 20× magnification. Control (**a**,**f**), BSA (**b**,**g**), 0.2 mM PA (**c**,**h**), 0.4 mM PA (**d**,**i**), and 0.8 mM PA (**e**,**j**). Quantification of lipids in (**H**) AML12 and (**I**) MIHA using ORO staining. The data were presented as means ± SD for three replicates. Data were analyzed with a one-way ANOVA followed by group comparisons using a post hoc Tukey’s multiple comparison test. Abbreviation: OS, overall survival; PPS, post-progression survival; HR, hazard ratio; PA, palmitic acid.

**Figure 2 cells-12-02311-f002:**
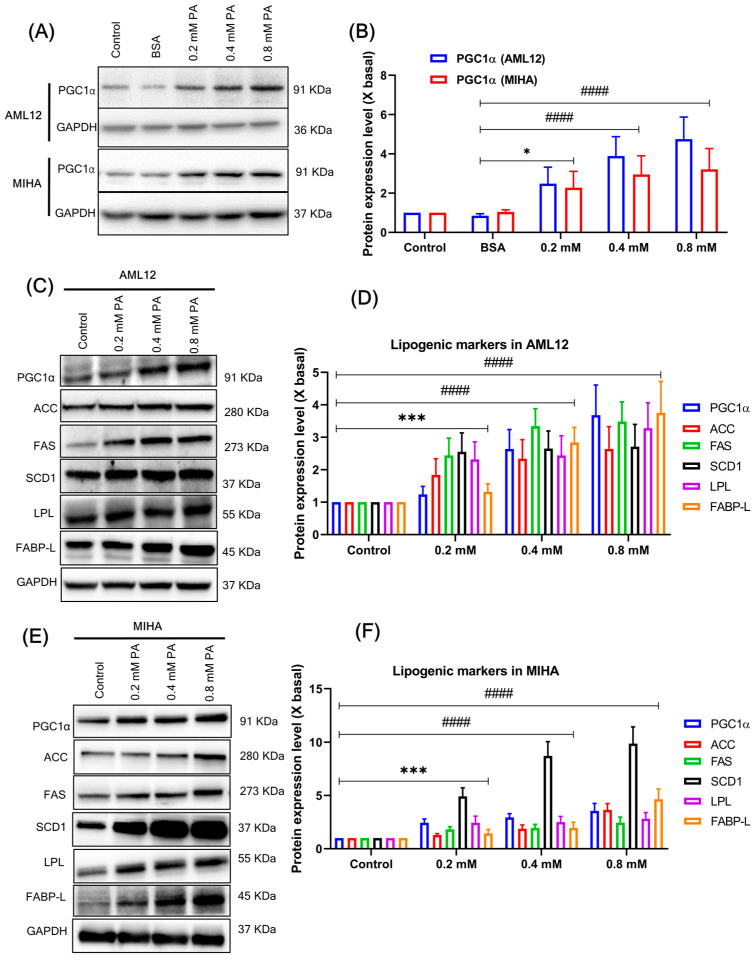
(**A**) Expression of PGC1α by Western Blot analysis. The respective hepatocytes were treated with PA (0.2, 0.4, and 0.8 mM) for 48 h, and the expression of PGC1α was dose-dependent. (**B**) The quantification of the target protein was calculated with a densitometer. The values are expressed as the fold of change (X basal), in mean ± SEM, where *n* = 3. * *p* < 0.01 and #### *p* < 0.0001 was considered a significant result compared to BSA control. (**C**) Representative western blot results showed PA-induced expression of lipogenic markers. AML12 cells were treated with PA (0.2, 0.4, and 0.8 mM) for 24 h, and the expression of lipogenic markers was dose-dependent. (**D**) The quantification of the target protein was calculated with a densitometer. The values are expressed as the fold of change (X basal), in Mean ± SEM, where *n* = 3. *** *p* < 0.01 and #### *p* < 0.0001 compared to the control. (**E**) Representative western blot results showing PA-induced expression of lipogenic markers. MIHA cells were treated with PA (0.2, 0.4, and 0.8 mM) for 24 h, and the expression of lipogenic markers was dose-dependent. (**F**) The quantification of the target protein was calculated with a densitometer. The values are expressed as the fold of change (X basal), in Mean ± SEM, where *n* = 3. *** *p* < 0.01 and #### *p* < 0.0001 compared to the control. Abbreviation: PGC1α, peroxisome proliferator-activated receptor gamma coactivator 1 alpha; PA, palmitic acid; GAPDH, Glyceraldehyde 3-phosphate dehydrogenase; ACC, acetyl-CoA carboxylase; FAS, fatty acid synthase; SCD1, stearoyl CoA desaturase-1; LPL, lipoprotein lipase; FABP-L, liver-type fatty acid-binding protein.

**Figure 3 cells-12-02311-f003:**
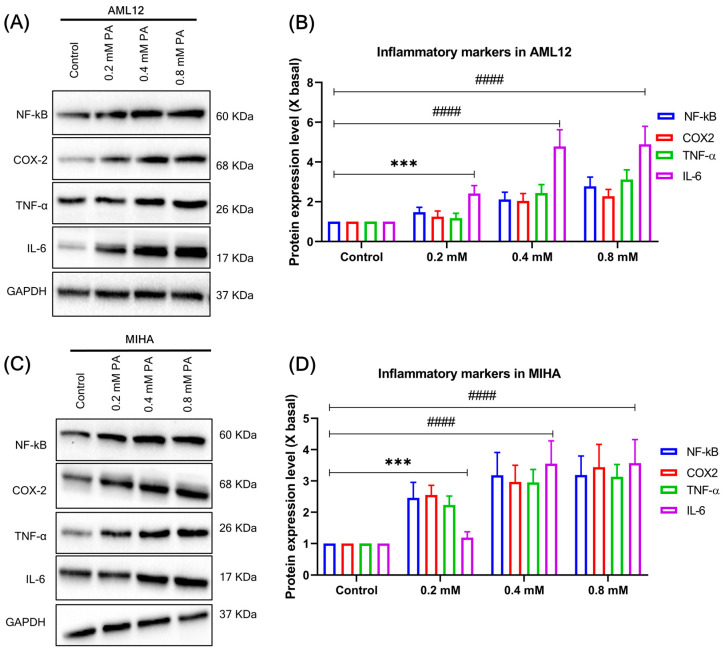
(**A**) Representative western blot results showed PA-induced expression of inflammatory markers. AML12 cells were treated with PA (0.2, 0.4, 0.8 mM) for 24 h, and the expression of inflammatory markers was dose-dependent. (**B**) The quantification of the target protein was calculated with a densitometer. The values are expressed as the fold of change (X basal), in mean ± SEM, where *n* = 3. *** *p* < 0.001 and #### *p* < 0.0001 compared to the control. (**C**) Representative western blot results showed PA-induced expression of inflammatory markers. MIHA cells were treated with PA (0.2, 04, 0.8 mM) for 24 h, and the expression of inflammatory markers was dose-dependent. (**D**) The quantification of the target protein was calculated with a densitometer. The values are expressed as the fold of change (X basal), in Mean ± SEM, where *n* = 3. *** *p* < 0.001 and #### *p* < 0.0001 compared to the control. Abbreviation: PA, palmitic acid; NF-kB, nuclear factor-kappa B; COX-2, cyclooxygenase-2; TNF-α, tumor necrosis factor-alpha; IL-6, interleukin-6; GAPDH, Glyceraldehyde 3-phosphate dehydrogenase.

**Figure 4 cells-12-02311-f004:**
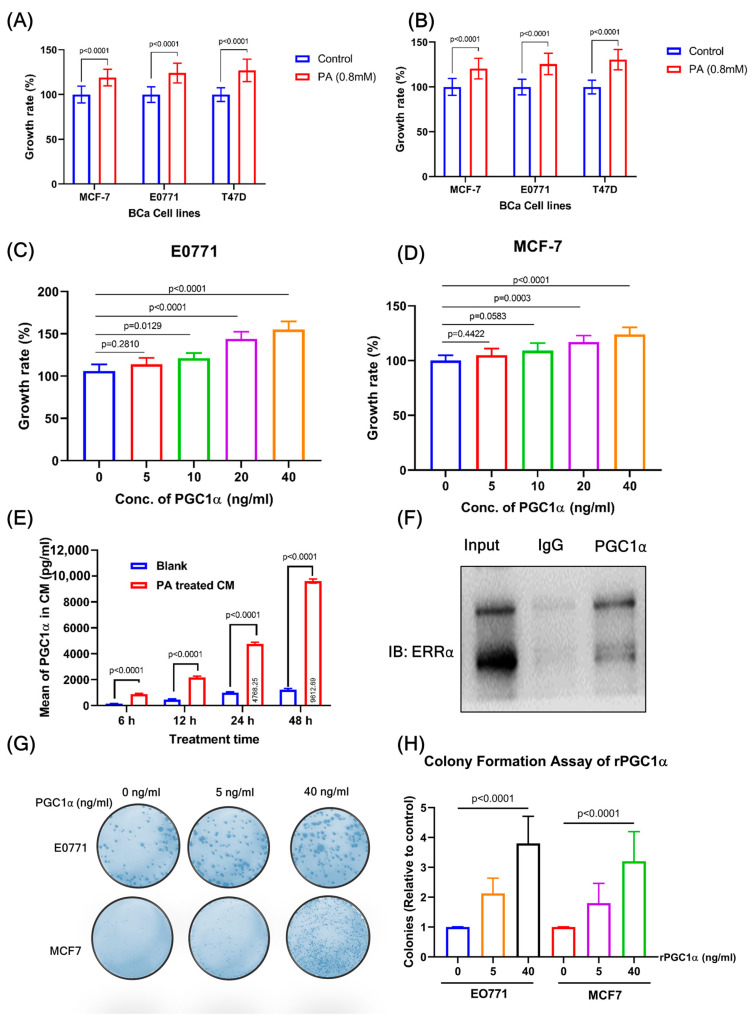
CM derived from (**A**) AML12 cells and (**B**) MIHA cells promote the proliferation of BC. The respective BC cell lines were treated with conditional medium obtained from hepatic cells for 24 h, followed by the assay of MTT. The data are presented as means ± SD for three replicates. Data were analyzed with a student’s *t*-test. CM was prepared from hepatic cells that were previously treated with PA (0.8 mM) for 24 h, followed by 12 h treatment with fresh complete medium. In the next step, CM was filtered and diluted with 25% fresh medium. The values are expressed in mean ± SEM, where *n* = 3. (**C**,**D**) Treatment of rPGC1α promotes cell proliferation in BC (E0771 and MCF7) cell lines. The respective BC cell lines were treated with PGC1α (0–40 ng/mL) for 24 h, followed by an MTT assay. The data are presented as means ± SD for three replicates. Data were analyzed with a one-way ANOVA followed by group comparisons using a post hoc Tukey’s multiple comparison test. (**E**) Quantification of PGC1α in CM using ELISA. CM was prepared from hepatic cells that were previously treated with 0.8 mM PA for 6, 24, and 48 h, followed by a 12 h treatment with a fresh complete medium. Elevated levels of PGC1α were observed in the CM at all time points compared to those in the control, according to ELISA results. The values are given in mean SEM, with *n* = 3. (**F**) The results of the co-immunoprecipitation assay suggested an interaction between PGC1α and ERRα. (**G**) PGC1α promotes colony formation in BC cells. Clonogenic assay results for E0771 and MCF7 cells treated with rPGC1α at the indicated concentrations. The data are presented as means ± SD for three replicates. Data were analyzed with a one-way ANOVA followed by group comparisons using a post hoc Tukey’s multiple comparison test. (**H**) The clonogenic number was obtained from the treatment of rPGC1α with the indicated concentrations in E0771 and MCF7 cells. The colonies’ numbers were counted. The data are presented as means ± SD for three replicates. Data were analyzed with a one-way ANOVA followed by group comparisons using a post hoc Tukey’s multiple comparison test.

**Figure 5 cells-12-02311-f005:**
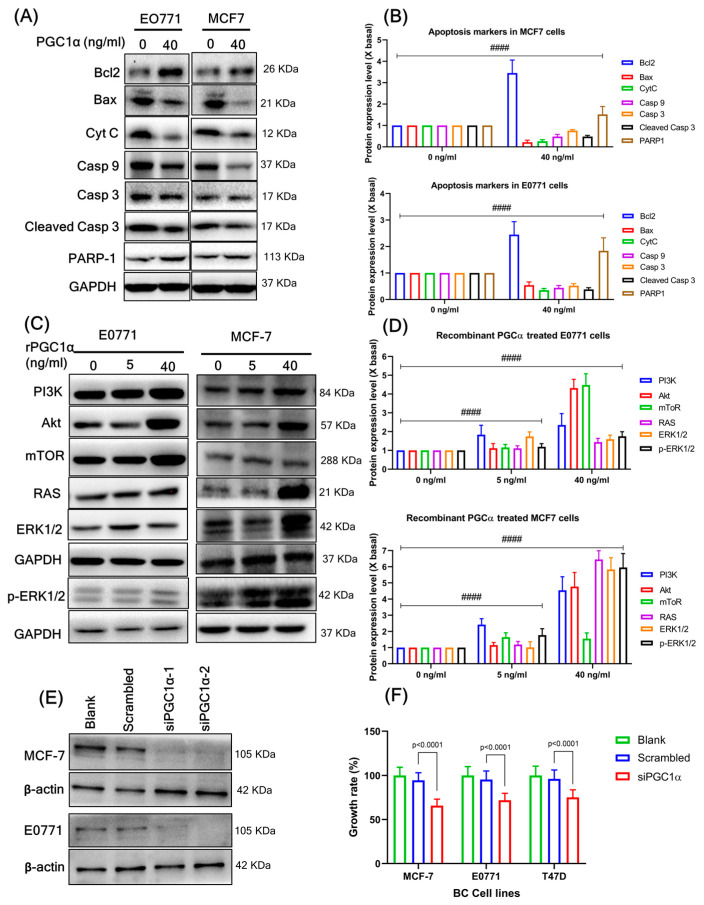
(**A**) High levels of rPGC1α inhibit apoptosis-related factors. The respective BC (E0771 and MCF7) cell lines were treated with PGC1α (5 and 40 ng/mL) for 24 h followed by the assay of apoptosis markers using western blot. (**B**) The quantification of the target protein was calculated with a densitometer. The values are expressed as the fold of change (X basal), in mean ± SEM, where *n* = 3. #### *p* < 0.0001 was considered a significant result when compared to the untreated rPGC1α group. (**C**) Protein expression of rPGC1α treated BC cell lines. Treatment of rPGC1α (0, 5, and 40 ng/mL) increased cell proliferation through the downstream expression of RAS-RAF-MAPK and PI3K-Akt-mTOR pathways on respective BC cell lines. (**D**) The quantification of the target protein was calculated with a densitometer. The values are expressed as the fold of change (X basal), in mean ± SEM, where *n* = 3. #### *p* < 0.0001 was considered a significant result when compared to the untreated rPGC1α group. (**E**) Deletion of the PGC1 gene in BC cells by transfection using Lipofectamine RNAiMAX. Represents the two sets of PGC1α siRNA transfection and thereby the absence of PGC1α expression when compared to a blank and negative control, which were analyzed by western blot. (**F**) Deletion of the PGC1α gene in BC cells for 24 h decreased the growth rate, which was determined by the MTT assay. The data are presented as means ± SEM for three replicates. Data were analyzed with one-way ANOVA followed by group comparisons using a post hoc Tukey’s multiple comparison test. Abbreviation: Bcl2, B-cell lymphoma 2; BAX, Bcl-2 Associated X-protein; cyc, cytochrome; Casp, caspase; PARP1, Poly [ADP-ribose] polymerase 1; PI3K, Phosphoinositide 3-kinases; Akt, serine/threonine-protein kinase; mTOR, mammalian target of rapamycin; RAS, rat sarcoma; ERK1/2, extracellular signal-regulated kinase 2; GAPDH, Glyceraldehyde 3-phosphate dehydrogenase.

**Figure 6 cells-12-02311-f006:**
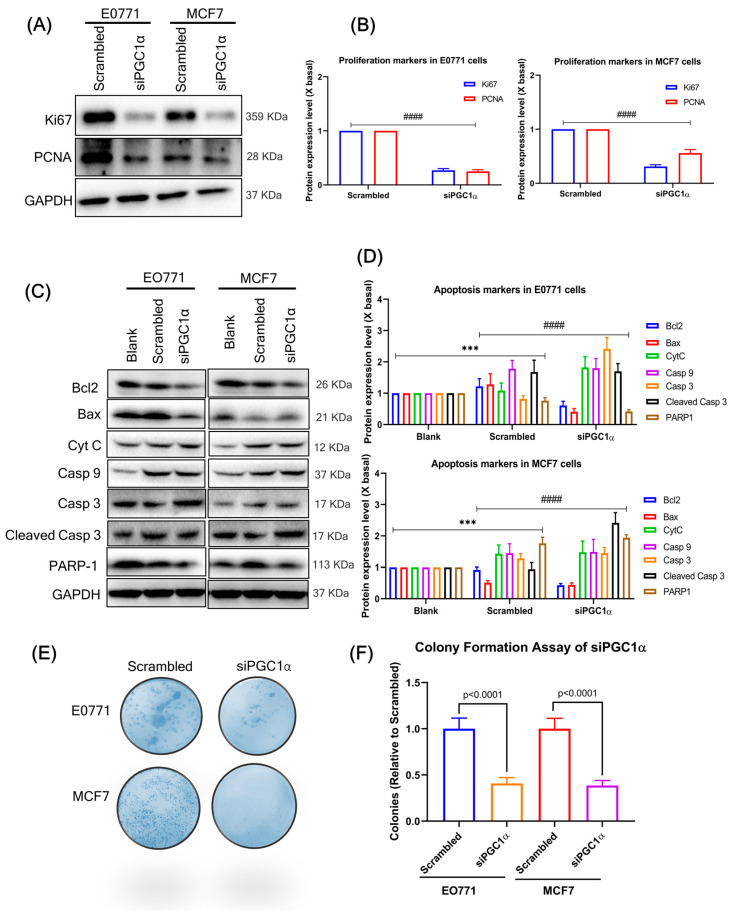
(**A**) Silencing of PGC1α inhibits BC cell proliferation. Scrambled or siPGC1α of respective BC cells treated with the medium for 24 h followed by the assay of proliferation markers (Ki67 and PCNA) using a western blot. (**B**) The quantification of the target protein was calculated with a densitometer. The values are expressed as the fold of change (X basal), in mean ± SEM, where *n* = 3. #### *p* < 0.0001 was considered a significant result when compared to the scrambled group. (**C**) Silencing of PGC1α promotes BC cell apoptosis-related factors. Scrambled or siPGC1α of respective BC cells treated with the medium for 24 h, followed by the assay of apoptosis markers using a western blot. (**D**) The quantification of the apoptosis target protein was calculated with a densitometer. The values are expressed as the fold of change (X basal), in mean ± SEM, where *n* = 3. *** *p* < 0.01 was considered a significant result when compared to the control, and #### *p* < 0.001 was considered a significant result when compared to the scrambled group. (**E**) siPGC1α inhibits colony formation in BC cells. Clonogenic assay results for E0771 and MCF7 cells treated with scrambled or siPGC1α. (**F**) Quantification of colonies treated with scrambled or siPGC1α and colonies counted relative to scrambled. The data are presented as means ± SD for three replicates. Data were analyzed with a one-way ANOVA followed by group comparisons using a post hoc Tukey’s multiple comparison test. Abbreviation: PCNA, Proliferating cell nuclear antigen; Bcl2, B-cell lymphoma 2; BAX, Bcl-2 Associated X-protein; cyc, cytochrome; Casp, caspase; PARP1, Poly [ADP-ribose] polymerase 1; GAPDH, Glyceraldehyde 3-phosphate dehydrogenase.

**Figure 7 cells-12-02311-f007:**
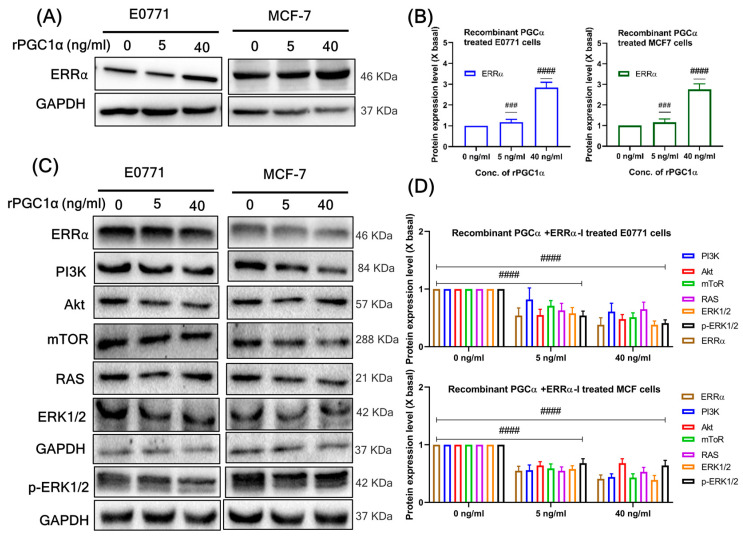
(**A**) Expression of ERRα in the rPGC1α-treated BC cells by Western blot analysis. The respective BC cells were injected with rPGC1α (0, 5, 40 ng/mL) for 24 h and the expression of ERRα was dose-dependent. (**B**) The quantification of the target protein was calculated with a densitometer. The values are expressed as the fold of change (X basal), in mean ± SEM, where *n* = 3. ### *p* < 0.01 and #### *p* < 0.0001 was considered a significant result when compared to the untreated rPGC1α group. (**C**) ERRα antagonist (XCT-790) inhibits PGC1α-induced BC proliferation. The respective BC cells were treated with XCT-790 (5 μM) and rPGC1α (0, 5, 40 ng/mL) for 24 h, which reduced the expression of ERRα cell proliferation markers through the downstream expression of RAS-RAF-MAPK and PI3K-Akt-mTOR pathways on respective BC cell lines. (**D**) The quantification of the target protein was calculated with a densitometer. The values are expressed as the fold of change (X basal), in mean ± SEM, where *n* = 3. #### *p* < 0.0001 was considered a significant result when compared to the untreated rPGC1α group. Abbreviation: ERRα, estrogen-related receptor alpha; rPGC1α, recombinant peroxisome proliferator-activated receptor gamma coactivator 1 alpha; PI3K, Phosphoinositide 3-kinases; Akt, serine/threonine-protein kinase; mTOR, mammalian target of rapamycin; RAS, rat sarcoma; ERK1/2, extracellular signal-regulated kinase 2; GAPDH, Glyceraldehyde 3-phosphate dehydrogenase.

**Figure 8 cells-12-02311-f008:**
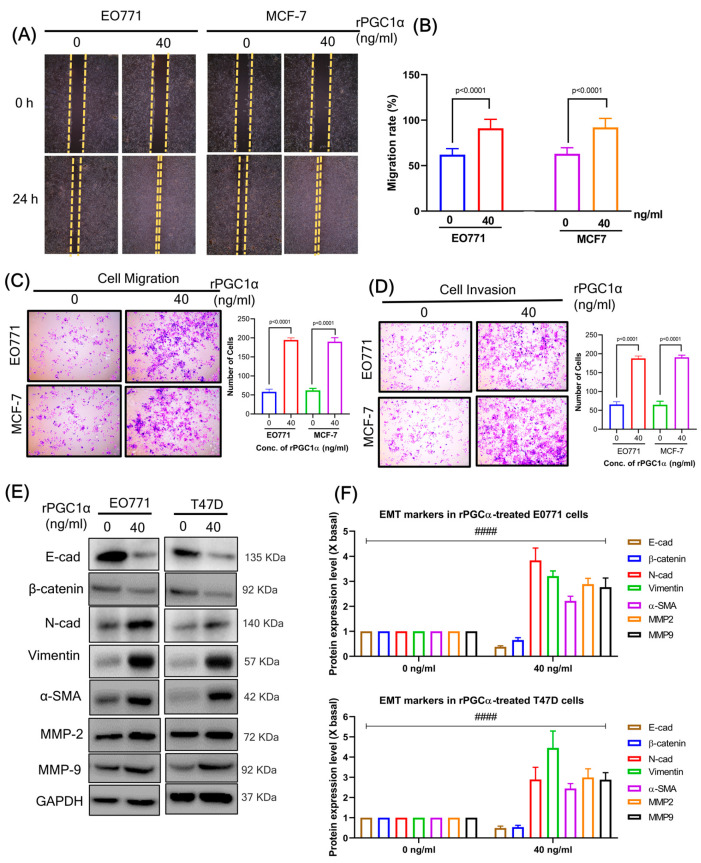
(**A**) A high level of PGC1α promotes migration. The effect of PGC1α on cell motility was analyzed by a wound-healing assay. (**B**) The data were analyzed with a one-way ANOVA followed by group comparisons using a post hoc Tukey’s multiple comparison test. (**C**) A high level of PGC1α promotes migration. The effect of rPGC1α on cell motility was examined by a transwell assay. The data are presented as means ± SD for three replicates. Data were analyzed with a one-way ANOVA followed by group comparisons using a post hoc Tukey’s multiple comparison test. (**D**) A high level of PGC1α promotes invasion. The effect of rPGC1α on cell motility was examined by a matrigel-coated transwell assay. The data are presented as means ± SD for three replicates. Data were analyzed with a one-way ANOVA followed by group comparisons using a post hoc Tukey’s multiple comparison test. (**E**) A high level of PGC1α promotes the EMT process. The markers of EMT were determined using high levels of PGC1α on respective breast cancer cells (EO771 and T47D), which were measured by western blot. (**F**) The quantification of the target protein was calculated with a densitometer. The values are expressed as the fold of change (X basal), in mean ± SEM, where *n* = 3. #### *p* < 0.0001 was considered a significant result when compared to the untreated rPGC1α group. Abbreviation: rPGC1α, recombinant peroxisome proliferator-activated receptor gamma coactivator 1 alpha; E-cad; E-cadherin; N-cad, N-cadherin; α-SMA, alpha-smooth muscle actin; MMP, matrix metalloproteinases; GAPDH, Glyceraldehyde 3-phosphate dehydrogenase.

**Figure 9 cells-12-02311-f009:**
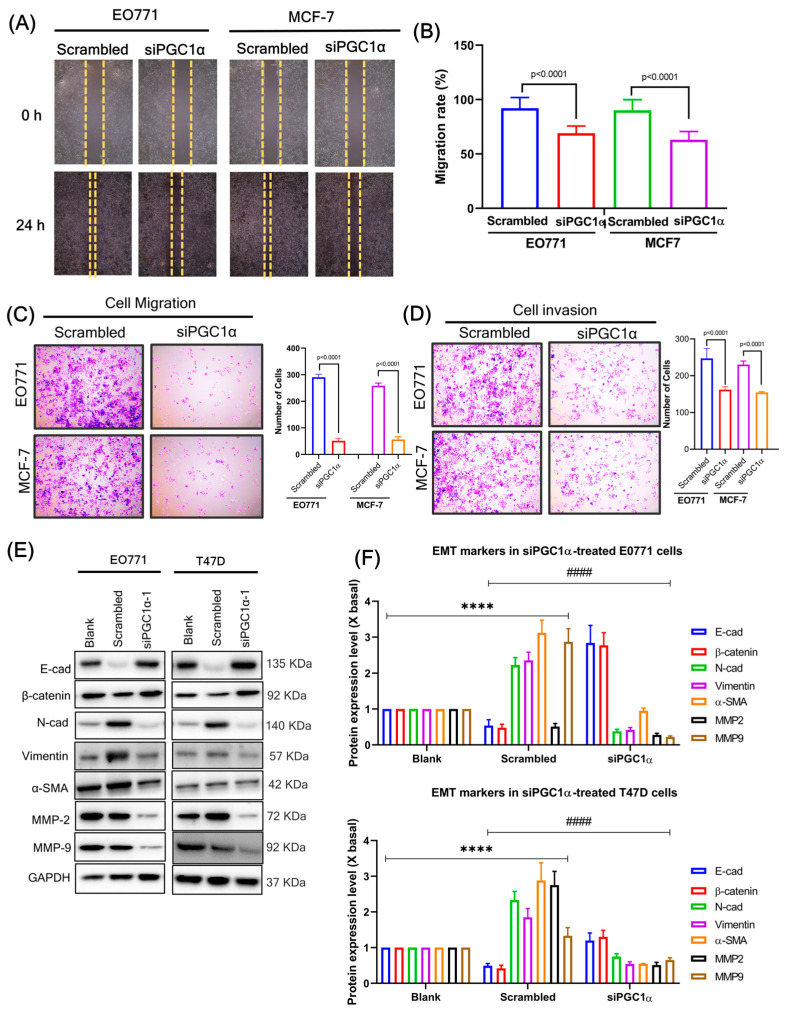
(**A**) Silencing of PGC1α suppresses migration. The effect of scrambled or siPGC1α on cell motility was analyzed by wound healing assay. (**B**) The data were analyzed with a one-way ANOVA followed by group comparisons using a post hoc Tukey’s multiple comparison test. (**C**) Silencing of PGC1α suppresses migration. The effect of PGC1α on cell motility was examined by transwell assay. The data are presented as means ± SD for three replicates. Data were analyzed with a one-way ANOVA followed by group comparisons using a post hoc Tukey’s multiple comparison test. (**D**) The silencing of PGC1α suppresses invasion. The effect of PGC1α on cell motility was assessed by a matrigel-coated transwell assay. The data are presented as means ± SD for three replicates. Data were analyzed with one-way ANOVA followed by group comparisons using a post hoc Tukey’s multiple comparison test. (**E**) Silencing of PGC1α suppresses the EMT process. As a result of treatment with the medium for 24 h, scrambled or siPGC1α cells of BC cells (EO771 and T47D) were analyzed by western blotting. (**F**) The quantification of the target protein was calculated with a densitometer. The values are expressed as the fold of change (X basal), in mean ± SEM, where *n* = 3. **** *p* < 0.0001 was considered a significant result when compared to blank, and #### *p <* 0.0001 was considered a significant result when compared to scrambled. Abbreviation: rPGC1α, recombinant peroxisome proliferator-activated receptor gamma coactivator 1 alpha; E-cad; E-cadherin; N-cad, N-cadherin; α-SMA, alpha-smooth muscle actin; MMP, matrix metalloproteinases; GAPDH, Glyceraldehyde 3-phosphate dehydrogenase.

## Data Availability

The corresponding author can provide the data for this article upon reasonable request.

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
