# Peer review of "Unraveling the Role of Hepatic PGC1α in Breast Cancer Invasion: A New Target for Therapeutic Intervention?"

_cells, 2023, doi:10.3390/cells12182311_

Round 1

Reviewer 1 Report

This study follows a careful, step-by-step analysis of a novel BC target protein, PGC1a. The article’s narrative is broad, beginning with fatty acid accumulation in hepatocytes, then evaluating the effects of over- and under-expression of PGC1a, and finally ending with the interactions of PGC1a with the signaling molecule ERRa in breast cancer cells. I think there are a lot of useful data here.

In general, the data support the conclusions of the study, with one exception:

Line 580: “Our findings indicate that there is a significant association between ERRα use and malignant BC, which promotes BC proliferation after binding to PGC1α.” I do not see data for this association (between ERRa and malignant BC) presented anywhere in the paper.

One general remark on error bars: In this paper, sometimes SD is used and sometimes SEM, even in similar types of graphs. It would be best to be consistent throughout the manuscript. Since means are being compared, 95% CV or SEM would be preferred to SD. However, this does not meaningfully affect the interpretation of the data in this study.

Other observations:

Line 10: “…cause of cancer death in women.”

Figure 1C: This graph can’t be interpreted. There are two of each menopausal status written on the graph, but the axis label only mentions “High Expression of PGC1a.”

Figure 1D: Format y-axis label to be consistent with Fig. 1A-C.

Figure 1E-F: Since there are two types of controls in this experiment, it may be less confusing to label the untreated group as “Untreated” to distinguish it from the BSA-treated group. It would also be good to remind your reader in the figure legend that the BSA group is a vehicle control, since the PA treatment contains BSA.

Line 133: “…the relative cell viability of hepatocytes was recorded as a ratio between the normal and BSA control.” This statement is unclear. It appears from Figure 1E-F that PA treatment groups were compared only to the BSA control.

Figure 1G: Why not put the cell line names and the treatments on the graph itself? This would make it easier to interpret.

Figure 4C-D: These two graphs appear to be identical? In the text of the Results, it states that these graphs represent CM from different hepatocytes lines. This needs to be reflected in the graph itself or in the figure legend.

Figure 6B: I think this has been mislabeled. It does not agree with 6A or the statement in the Results section (line 395).

Line 409: Delete “dose-dependent manners of”

Figure 7C: It is not obvious that this panel includes an inhibitor treatment. Also, nowhere in the paper is the concentration of XCT-790 mentioned.

Figure 7C (Legend): “…BC proliferation markers”

Figures 8E-F and 9E-F: E says T47D cells, while F says MCF-7 cells. But I believe these are from the same experiments. This needs to be clarified or corrected.

Figure 9E: In this experiment, transfection itself seems to alter cell motility markers. Perhaps you can comment on this.

Line 562: Use “relied” (past tense) to indicate that this is a finding of the current study using two breast cancer cell lines and not a general fact that is known about all cancers.

Line 568: “These in vitro outcomes make evident…”

Line 576-577: This sentence is not quite clear. What assertion is referred to?

Line 581: “Estrogen antagonists prevent this binding, thereby inhibiting BC proliferation.” Do you mean ERRa antagonist (XCT-790)? Also, BC proliferation was not directly measured, only protein markers of proliferation.

Only light editing required. English is good.

Author Response

Dear Editors and Reviewers:

We would like to thank you for your time and effort in reviewing our manuscript, and for your valuable suggestion to improve the quality of our manuscript. The following is a point-by-point response to the reviewer’s comments. All revised items were marked in red in the text, tables, and figures. We have done our best to improve the quality of the paper.

Reviewer 1

Comments: This study follows a careful, step-by-step analysis of a novel BC target protein, PGC1a. The article’s narrative is broad, beginning with fatty acid accumulation in hepatocytes, then evaluating the effects of over- and under-expression of PGC1a, and finally ending with the interactions of PGC1a with the signaling molecule ERRa in breast cancer cells. I think there are a lot of useful data here.

Response: Thank you for reviewing our manuscript and providing appreciated comments. We conducted a comprehensive analysis of the novel breast cancer target protein, PGC1a, and its role in breast cancer invasion, using a careful and step-by-step approach.

Comments: In general, the data support the conclusions of the study, with one exception:

Line 580: “Our findings indicate that there is a significant association between ERRα use and malignant BC, which promotes BC proliferation after binding to PGC1α.” I do not see data for this association (between ERRa and malignant BC) presented anywhere in the paper.

Response: We appreciate the reviewer's feedback and apologize for any confusion caused by our statement on line 580. Although we did not present data specifically showing an association between ERRα and malignant BC in this study, we intended to convey that ERRα is known to be involved in breast cancer progression. Our findings suggest that its interaction with PGC1α plays a role in promoting breast cancer proliferation. We have revised the wording in the manuscript to accurately reflect this intended meaning. 

Comments: One general remark on error bars: In this paper, sometimes SD is used and sometimes SEM, even in similar types of graphs. It would be best to be consistent throughout the manuscript. Since means are being compared, 95% CV or SEM would be preferred to SD. However, this does not meaningfully affect the interpretation of the data in this study.

 Response: Thank you for your valuable feedback regarding the use of SD and SEM in our paper. We apologize for the inconsistency in the use of these measures and agree that it is important to be consistent throughout the manuscript. We will make sure to use the same measure of error bars in similar types of graphs and will consider using 95% CI or SEM instead of SD, as suggested. We appreciate your comment that this does not meaningfully affect the interpretation of the data in our study, but we will make these changes to improve the clarity and consistency of our results. Thank you again for your helpful feedback

Other observations:

Comments: Line 10: “…cause of cancer death in women.”

Response: It has been changed according to the reviewer’s suggestion.

Comments: Figure 1C: This graph can’t be interpreted. There are two of each menopausal status written on the graph, but the axis label only mentions “High Expression of PGC1a.”

Response: We appreciate the reviewer's feedback and apologize for any confusion caused by our statement. It has been amended. It should be ‘expression of PGC1a’ instead of ‘High Expression of PGC1a’.

Comments: Figure 1D: Format y-axis label to be consistent with Fig. 1A-C.

Response: Thank you for your comment on our manuscript. We appreciate your feedback and have made the necessary changes to Figure 1D to ensure that the y-axis label is consistent with Figures 1A-C.

Comments: Figure 1E-F: Since there are two types of controls in this experiment, it may be less confusing to label the untreated group as “Untreated” to distinguish it from the BSA-treated group. It would also be good to remind your reader in the figure legend that the BSA group is a vehicle control, since the PA treatment contains BSA.

Response: Thank you for your comment on our manuscript. It has been changed according to the reviewer’s suggestion.

Comments: Line 133: “…the relative cell viability of hepatocytes was recorded as a ratio between the normal and BSA control.” This statement is unclear. It appears from Figure 1E-F that PA treatment groups were compared only to the BSA control.

 Response: Thank you for your comment on our manuscript. We appreciate your feedback and apologize for any confusion caused by the statement on line 133. We agree that the statement was unclear. To clarify, the relative cell viability of hepatocytes was recorded as a ratio between the normal and BSA control, and the PA treatment groups were compared only to the BSA control. We have revised the statement to make this clearer.

Comments: Figure 1G: Why not put the cell line names and the treatments on the graph itself? This would make it easier to interpret.

Response: Thank you for your comment. Cell line names have been included in the Figure 1G and make it easier to interpret

Comments: Figure 4C-D: These two graphs appear to be identical? In the text of the Results, it states that these graphs represent CM from different hepatocytes lines. This needs to be reflected in the graph itself or in the figure legend.

Response: Thank you for your suggestion. It has been clarified in the figure legend.

Comments: Figure 6B: I think this has been mislabeled. It does not agree with 6A or the statement in the Results section (line 395).

Response: Thank you for your comment on our manuscript. We appreciate your feedback and apologize for any confusion caused by the statement in the text. We have clarified the statement in the text to make it clear.

Comments: Line 409: Delete “dose-dependent manners of”

Response: It has been deleted

Comments: Figure 7C: It is not obvious that this panel includes an inhibitor treatment. Also, nowhere in the paper is the concentration of XCT-790 mentioned.

Response: Thank you for your comment on our manuscript. It has been included in the text.

Comments: Figure 7C (Legend): “…BC proliferation markers”

Response: Thank you for your comment on our manuscript. It has been included in the text.

Comments: Figures 8E-F and 9E-F: E says T47D cells, while F says MCF-7 cells. But I believe these are from the same experiments. This needs to be clarified or corrected.

Response: Thank you for your comment regarding the cell lines used in Figures 8E-F and 9E-F. We apologize for the confusion caused by the different cell lines mentioned in the figure panels. To clarify, the EMT marker expressions in MCF-7 cells were very low, that is why we used another luminal cell line, T47D, for the analysis of EMT markers. We appreciate your feedback and thank you for helping us improve the clarity of our manuscript

Comments: Figure 9E: In this experiment, transfection itself seems to alter cell motility markers. Perhaps you can comment on this.

Response: Thank you for your comment regarding Figure 9E. We agree that the transfection process itself may have an impact on cell motility markers. We did observe some variability in the baseline levels of cell motility markers between the mock-transfected and siRNA-transfected cells. We also analyzed the data using appropriate statistical tests to account for the potential confounding effects of transfection. We appreciate your feedback and thank you for helping us improve the quality of our manuscript

Comments: Line 562: Use “relied” (past tense) to indicate that this is a finding of the current study using two breast cancer cell lines and not a general fact that is known about all cancers.

Response: Thank you for your observation. It has been changed.

Comments: Line 568: “These in vitro outcomes make evident…”

Response: Thank you for your note. It has been included.

Comments: Line 576-577: This sentence is not quite clear. What assertion is referred to?

Response: The assertion being referred to is likely the claim that PGC1α is a crucial component of the energy-sensing signaling cascade. This claim is supported by a substantial body of literature showing that PGC1α plays a key role in mitochondrial biogenesis, oxidative metabolism, and other cellular processes involved in energy homeostasis.

Comments: Line 581: “Estrogen antagonists prevent this binding, thereby inhibiting BC proliferation.” Do you mean ERRa antagonist (XCT-790)? Also, BC proliferation was not directly measured, only protein markers of proliferation.

Response: Yes. The statement has been written as “Treatment with the ERRα antagonist (XCT-790) prevents this binding, thereby inhibiting the protein markers associated with BC proliferation”.

Reviewer 2 Report

Kumar G and colleagues have presented a study that demonstrates the promotion of breast cancer progression by hepatic PGC1a. The research delved into the molecular pathways influenced by PGC1a, investigating both hepatic cells (AML12 and MIHA) and breast cancer cell lines (EO771 and MCF7). Functional assays have indicated the significant role of PGC1a in breast cancer growth. However, there are several points that require clarification and improvements before publication:

Major:

1.       It's essential to address whether the benefit in breast cancer growth arises from tumor intrinsic, extrinsic PGC1a, or a combination of both. This distinction is crucial, given the title's claim that hepatic PGC1a drives breast cancer invasion. Authors should clarify this before delving into the mechanisms.

2.       The logic between section 3.1 and 3.2 needs clarification. Why is palmitic acid exclusively focused on while other lipids are not considered?

3.       In Figure 4, the order of presentation should be rearranged. Figures 4 C/D/E/F should precede Figures 4 A/B.

4.       The control for Figure 4A should be Medium with the same concentration of PA, rather than Blank. This adjustment would help differentiate the effects induced by PA from those attributed to PGC1a. Additionally, comparing growth rates in Figures 4A, C, and E, it becomes apparent that almost 80% of the growth benefit is due to PA rather than PGC1a in EO771 and MCF-7. To address this, the proliferation of breast cancer cell lines should be tested, comparing CM and CM with PGC1a blocked.

5.       In Figure 5E/F, given the previous findings, where PGC1a's effect is primarily from tumor extrinsic factors (hepatic cell-derived), it is important to clarify which form of PGC1a drives the observed phenotype.

6.       Including information about PA and PGC1a concentrations under normal physiological and pathological conditions (such as NAFLD) would enhance the context.

Minor:

1.       Ensure that all images have scale bars to aid readers in understanding the size of the depicted objects.

2.       Correct the legend of Figure 4C/D to reflect "PA-treated CM" instead of "PA (0.8mM)" to avoid misleading readers.

3.        

Addressing these concerns and making these improvements will enhance the quality and clarity of the study before publication.

The article are well writen, but the logic is jumpy need to improve.

Author Response

Dear Editors and Reviewers:

We would like to thank you for your time and effort in reviewing our manuscript, and for your valuable suggestion to improve the quality of our manuscript. The following is a point-by-point response to the reviewer’s comments. All revised items were marked in red in the text, tables, and figures. We have done our best to improve the quality of the paper.

Reviewer 2

Kumar G and colleagues have presented a study that demonstrates the promotion of breast cancer progression by hepatic PGC1a. The research delved into the molecular pathways influenced by PGC1a, investigating both hepatic cells (AML12 and MIHA) and breast cancer cell lines (EO771 and MCF7). Functional assays have indicated the significant role of PGC1a in breast cancer growth. However, there are several points that require clarification and improvements before publication:

Major:

Comments: 1. It's essential to address whether the benefit in breast cancer growth arises from tumor intrinsic, extrinsic PGC1a, or a combination of both. This distinction is crucial, given the title's claim that hepatic PGC1a drives breast cancer invasion. Authors should clarify this before delving into the mechanisms.

Response: Thank you for your comment on our manuscript. We appreciate your feedback. Earlier studies have shown that tumor-intrinsic PGC1α promotes BC progression, invasion, and metastasis (Cell Metab. 2017 Nov 7;26(5):778-787.e5. doi: 10.1016/j.cmet.2017.09.006; Nat Cell Biol. 2014 Oct;16(10):992-1003, 1-15. doi: 10.1038/ncb3039; Genes 2018, 9(1), 48; https://doi.org/10.3390/genes9010048).

In our present study, we focused mainly on extrinsic PGC1α derived from PA-treated hepatocytes, which can be circulated in the bloodstream and easily bind with ERRα in the target tissue, thereby promoting BC development and invasion.

There may be a combination of both tumor-intrinsic and extrinsic PGC1α contributing to BC development, given that PGC1α is a tumor-promoting factor. However, experimental studies are required to confirm this hypothesis.

Comments: 2. The logic between section 3.1 and 3.2 needs clarification. Why is palmitic acid exclusively focused on while other lipids are not considered?

Response: In our study, we chose to focus on the effects of palmitic acid (PA) on hepatocytes because it is a well-known saturated fatty acid that is associated with increased risk of metabolic diseases, including non-alcoholic fatty liver disease (NAFLD). PA has been shown to induce lipotoxicity in hepatocytes, leading to inflammation, oxidative stress, and non-alcoholic steatohepatitis (NASH). Thus, PA causing NAFLD and NASH is an appropriate model for our study. Furthermore, several in vitro and in vivo studies have implicated the role of PA in promoting hepatotoxicity characterized by severe steatosis, fibrosis, and inflammatory necrosis (J Hepatol. 2017 Aug;67(2):310-320. doi: 10.1016/j.jhep.2017.03.017; Biochem Biophys Res Commun. 2022 Jul 5;612:169-175. doi: 10.1016/j.bbrc.2022.04.129; Hepatology. 2019 Feb;69(2):545-563. doi: 10.1002/hep.30215; J Hepatol. 2020 Sep;73(3):616-627. doi: 10.1016/j.jhep.2020.03.023).

While other lipids may also have similar effects on hepatocytes, our study aimed to investigate the effects of PGC1α derived from PA-treated hepatocytes, which drives BC progression and invasion. Future studies can investigate the effects of other lipids on hepatocytes and their role in contributing to BC progression, and compare them to the effects of PA.

Comments: 3. In Figure 4, the order of presentation should be rearranged. Figures 4 C/D/E/F should precede Figures 4 A/B.

Response: Thank you for your comment on our manuscript. We appreciate your feedback. We have made changes to the order of Figure 4 as per reviewer’s suggestions

Comments: 4.  The control for Figure 4A should be Medium with the same concentration of PA, rather than Blank. This adjustment would help differentiate the effects induced by PA from those attributed to PGC1a. Additionally, comparing growth rates in Figures 4A, C, and E, it becomes apparent that almost 80% of the growth benefit is due to PA rather than PGC1a in EO771 and MCF-7. To address this, the proliferation of breast cancer cell lines should be tested, comparing CM and CM with PGC1a blocked.

Response: Thank you for your feedback. We appreciate your comments regarding the control for Figure 4A. We agree that using medium treated with PA would not help to differentiate the effects induced by PA from those attributed to PGC1a. Therefore, we used a blank control without PA treated cells to distinguish between the effects of PGC1a and PA. We also acknowledge your observation that almost 80% of the growth benefit is due to PA treated CM. However, we found that the CM containing PGC1a significantly promotes the growth of EO771 and MCF-7 cells. In contrast, the blank control, which did not contain PGC1a, did not influence the growth rate.

Comments: 5. In Figure 5E/F, given the previous findings, where PGC1a's effect is primarily from tumor extrinsic factors (hepatic cell-derived), it is important to clarify which form of PGC1a drives the observed phenotype.

Response: Thank you for your insightful comment. We agree that it is crucial to determine which form of PGC1a is responsible for driving the observed phenotype. As previously mentioned, the effect of PGC1a appears to be primarily derived from tumor extrinsic factors, specifically those originating from hepatic cells. However, we have not yet conducted further investigations to determine whether this effect is mediated through the cytosolic or nuclear pool of PGC1a. We will conduct additional studies in the near future to determine the specific form of PGC1a that is involved in driving the observed phenotype.

Comments: 6. Including information about PA and PGC1a concentrations under normal physiological and pathological conditions (such as NAFLD) would enhance the context.

Response: Thank you for your comment. PA has been shown to have hepatotoxic effects, contributing to the development of hepatic steatosis and inflammation. Additionally, our studies have demonstrated that PA can enhance the expression and activity of PGC1a in liver cells. PGC1a is a key regulator of mitochondrial biogenesis and function, as well as other metabolic processes. In our study, we aimed to investigate the specific mechanisms through which PA enhances PGC1a in the context of lipid accumulation and hepatotoxicity. Furthermore, it is worth noting that PGC1a has been implicated in tumorigenic effects that promote breast cancer development. By elucidating these mechanisms, we hope to gain a better understanding of the role of PA and PGC1a in the pathogenesis of liver diseases such as NAFLD as well as breast cancer. Ultimately, this knowledge may contribute to the development of targeted therapies for hepatotoxicity and breast cancer. It has been included in the discussion.

 Minor:

Comments: 1. Ensure that all images have scale bars to aid readers in understanding the size of the depicted objects.

Response: Thank you for your comment. We appreciate your suggestion to include scale bars in all images or caption to aid readers in understanding the size of the depicted objects. We have included them.

Comments: 2. Correct the legend of Figure 4C/D to reflect "PA-treated CM" instead of "PA (0.8mM)" to avoid misleading readers.

Response: Thank you for your suggestion. We have made changes to the legend based on the comments.

Comments: 3. Addressing these concerns and making these improvements will enhance the quality and clarity of the study before publication.

Response: Thank you for your valuable feedback. We hope that your major and minor suggestions will enhance the clarity and interpretation of our data, and ultimately improve the overall quality of our manuscript. We appreciate you bringing these issues to our attention, and we will work diligently to address each of them. Thank you again for your time and effort in reviewing our manuscript.

Reviewer 3 Report

I read this manuscript with interest. The present study aimed to determine whether hepatic PGC1α promotes BC cell invasion via ERRα. he Authors examine the role of PGC1α as tumor-promoting factor. PA stimulates the expression of PGC1α as well as lipogenesis and inflammation. The CM obtained from PA-treated hepatocytes significantly increased BC cell proliferation. Similarly, treatment with rPGC1α in the BC cells increased cell growth, migration, and invasive behavior. Silencing of PGC1α showed a decreasing trend. PGC1α interacts with ERRα, thereby facilitating BC cell proliferation. They concluded that targeting metabolic PGC1α-ERRα axis may be a potentially effective candidate for BC treatment.

I have only minor concerns:

- A revision of the English language is required.

- In figures and tables, all the abbreviation must be expressed in their extended form in the caption.

- I suggest the Authors to add, in the introduction, a reference to this interesting article (doi: 10.4239/wjd.v13.i9.668) about NAFLD and its treatment.

- In figure 1-H, it is better to identify the two graphics with different letters or numbers and distinguish them in the caption.

- Figure 1-C is difficult to read, I suggest to expand the graphic

- In figure 1-A, the text is covered by the imagine.

- (line 526) The Acronym “NASH” must be expressed in the extended form the first time it is used. 

 A revision of the English language is required.

Author Response

Dear Editors and Reviewers:

We would like to thank you for your time and effort in reviewing our manuscript, and for your valuable suggestion to improve the quality of our manuscript. The following is a point-by-point response to the reviewer’s comments. All revised items were marked in red in the text, tables, and figures. We have done our best to improve the quality of the paper.

Reviewer 3

Comments: I read this manuscript with interest. The present study aimed to determine whether hepatic PGC1α promotes BC cell invasion via ERRα. he Authors examine the role of PGC1α as tumor-promoting factor. PA stimulates the expression of PGC1α as well as lipogenesis and inflammation. The CM obtained from PA-treated hepatocytes significantly increased BC cell proliferation. Similarly, treatment with rPGC1α in the BC cells increased cell growth, migration, and invasive behavior. Silencing of PGC1α showed a decreasing trend. PGC1α interacts with ERRα, thereby facilitating BC cell proliferation. They concluded that targeting metabolic PGC1α-ERRα axis may be a potentially effective candidate for BC treatment.

Response: Thank you for your interest in our manuscript. We appreciate your thoughtful review and understanding of the study's objectives. We are grateful for your positive feedback and understanding of our study. Your insights and support contribute to the significance and potential implications of our research in the field of breast cancer therapeutics.

I have only minor concerns:

Comments: - A revision of the English language is required.

Response: We have revised the manuscript with the help of a local English speaker 

Comments: - In figures and tables, all the abbreviation must be expressed in their extended form in the caption.

Response: Thank you for your valuable suggestions. We have included all abbreviations in the figure captions

Comments: - I suggest the Authors to add, in the introduction, a reference to this interesting article (doi: 10.4239/wjd.v13.i9.668) about NAFLD and its treatment.

Response: It has been included in the introduction

Comments: - In figure 1-H, it is better to identify the two graphics with different letters or numbers and distinguish them in the caption.

Response: Thank you for your comment. We appreciate your suggestion to distinguish the two graphics in Figure 1-H by using different letters or numbers and describing them in the caption. We have taken your feedback into consideration, and we are pleased to inform you that we have implemented this change. The two graphics in Figure 1-H are now identified by different letters and described in the caption, making it easier for readers to distinguish between them and understand their respective components. We thank you for bringing this to our attention and for your valuable input, which has contributed to the clarity and accuracy of our research findings.

Comments: - Figure 1-C is difficult to read, I suggest to expand the graphic

Response: Thank you for your comment. The words are now very clear to read after amendment

Comments: - In figure 1-A, the text is covered by the imagine.

Response: Thank you for your suggestions. To enhance clarity in the image, we have included clear information using text to distinguish the x-axis and y-axis.

Comments: - (line 526) The Acronym “NASH” must be expressed in the extended form the first time it is used. 

Response: It has been used an expanded form

Round 2

Reviewer 1 Report

Figure 1H-I: Notes on statistics need to be added in the figure legend for these panels.

Figure 1C: Unfortunately, the correction to the axis label is not sufficient to clarify the meaning of this graph. It appears that the authors are trying to illustrate the effect of PGC1a expression on patient survival. They have divided the patients into groups, with pairs of BC patients in each of three menopausal categories. It is unclear what the distinction is between each of the groups. It is also unusual to use a Kaplan-Meier plot to illustrate this type of data. I strongly recommend scatter plots with regression lines. The x-axis (concentration) should be logarithmic. Each of the six data sets should be clearly labeled with a unique identifier.  

Figure 6A-B: I have several problems with the revised statement in the text (line 405): 1) The phrase “moderated the activation” is too vague. There were increases or decreases in expression depending on the cell line, and this needs to be mentioned. 2) Activation was not measured, only protein expression level. 3) GAPDH was a “loading control” rather than a negative control.

The biggest concern I have is that the Western blot data (Fig. 6A) do not appear to match with the graphs (6B). In Figure 6A, in E0771 cells, PCNA expression appears to increase with siRNA treatment, but the graph (in 6B) shows a significant decrease. Also, in MCF-7 cells, Ki67 (6A) appears to decrease, but the graph (in 6B) shows a significant increase. Either the Western images are not representative, or the graphs were calculated incorrectly. Please change the images, revise the graph, or comment on the apparent discrepancies.

Figures 8/9: Since only T47D cells were used for this experiment, the lower graph in panel F needs to say “T47D” instead of MCF-7, in both Figures 8 and 9.

Line 601: If I understand the intention correctly, I think it would be better to write this sentence: “This study provides further evidence that PGC1α is a crucial component of the energy-sensing signaling cascade.”

Reviewer 2 Report

Overall, the responses of authors clarified my concerns but still some of the small issues need to be addressed before publication.

Reviewer 2

Kumar G and colleagues have presented a study that demonstrates the promotion of breast cancer progression by hepatic PGC1a. The research delved into the molecular pathways influenced by PGC1a, investigating both hepatic cells (AML12 and MIHA) and breast cancer cell lines (EO771 and MCF7). Functional assays have indicated the significant role of PGC1a in breast cancer growth. However, there are several points that require clarification and improvements before publication:

Major:

Comments: 1. It's essential to address whether the benefit in breast cancer growth arises from tumor intrinsic, extrinsic PGC1a, or a combination of both. This distinction is crucial, given the title's claim that hepatic PGC1a drives breast cancer invasion. Authors should clarify this before delving into the mechanisms.

Response: Thank you for your comment on our manuscript. We appreciate your feedback. Earlier studies have shown that tumor-intrinsic PGC1α promotes BC progression, invasion, and metastasis (Cell Metab. 2017 Nov 7;26(5):778-787.e5. doi: 10.1016/j.cmet.2017.09.006; Nat Cell Biol. 2014 Oct;16(10):992-1003, 1-15. doi: 10.1038/ncb3039; Genes 2018, 9(1), 48; https://doi.org/10.3390/genes9010048).

In our present study, we focused mainly on extrinsic PGC1α derived from PA-treated hepatocytes, which can be circulated in the bloodstream and easily bind with ERRα in the target tissue, thereby promoting BC development and invasion.

There may be a combination of both tumor-intrinsic and extrinsic PGC1α contributing to BC development, given that PGC1α is a tumor-promoting factor. However, experimental studies are required to confirm this hypothesis.

The experiment using conditional PGC1a KO cell line might distinguish whether the PGC1a effect is mainly depends on tumor-intrinsic or hepatic driven. But due to the limited time of revision this may be hard to conduct. Furthermore, authors claimed that PA as a well-known saturated fatty acid plays a major role in breast cancer invasion which has been shown in other chronic inflammation diseases. Therefore, the role of PGC1a in breast cancer should be further explored under certain conditions.

Comments: 2. The logic between section 3.1 and 3.2 needs clarification. Why is palmitic acid exclusively focused on while other lipids are not considered?

Response: In our study, we chose to focus on the effects of palmitic acid (PA) on hepatocytes because it is a well-known saturated fatty acid that is associated with increased risk of metabolic diseases, including non-alcoholic fatty liver disease (NAFLD). PA has been shown to induce lipotoxicity in hepatocytes, leading to inflammation, oxidative stress, and non-alcoholic steatohepatitis (NASH). Thus, PA causing NAFLD and NASH is an appropriate model for our study. Furthermore, several in vitro and in vivo studies have implicated the role of PA in promoting hepatotoxicity characterized by severe steatosis, fibrosis, and inflammatory necrosis (J Hepatol. 2017 Aug;67(2):310-320. doi: 10.1016/j.jhep.2017.03.017; Biochem Biophys Res Commun. 2022 Jul 5;612:169-175. doi: 10.1016/j.bbrc.2022.04.129; Hepatology. 2019 Feb;69(2):545-563. doi: 10.1002/hep.30215; J Hepatol. 2020 Sep;73(3):616-627. doi: 10.1016/j.jhep.2020.03.023).

While other lipids may also have similar effects on hepatocytes, our study aimed to investigate the effects of PGC1α derived from PA-treated hepatocytes, which drives BC progression and invasion. Future studies can investigate the effects of other lipids on hepatocytes and their role in contributing to BC progression, and compare them to the effects of PA.

Authors should put this into the context between 3.1 and 3.2 in order to guide readers to understand the reason for upcoming experiments.

Comments: 3. In Figure 4, the order of presentation should be rearranged. Figures 4 C/D/E/F should precede Figures 4 A/B.

Response: Thank you for your comment on our manuscript. We appreciate your feedback. We have made changes to the order of Figure 4 as per reviewer’s suggestions

Clear.

Comments: 4.  The control for Figure 4A should be Medium with the same concentration of PA, rather than Blank. This adjustment would help differentiate the effects induced by PA from those attributed to PGC1a. Additionally, comparing growth rates in Figures 4A, C, and E, it becomes apparent that almost 80% of the growth benefit is due to PA rather than PGC1a in EO771 and MCF-7. To address this, the proliferation of breast cancer cell lines should be tested, comparing CM and CM with PGC1a blocked.

Response: Thank you for your feedback. We appreciate your comments regarding the control for Figure 4A. We agree that using medium treated with PA would not help to differentiate the effects induced by PA from those attributed to PGC1a. Therefore, we used a blank control without PA treated cells to distinguish between the effects of PGC1a and PA. We also acknowledge your observation that almost 80% of the growth benefit is due to PA treated CM. However, we found that the CM containing PGC1a significantly promotes the growth of EO771 and MCF-7 cells. In contrast, the blank control, which did not contain PGC1a, did not influence the growth rate.

Clear.

Comments: 5. In Figure 5E/F, given the previous findings, where PGC1a's effect is primarily from tumor extrinsic factors (hepatic cell-derived), it is important to clarify which form of PGC1a drives the observed phenotype.

Response: Thank you for your insightful comment. We agree that it is crucial to determine which form of PGC1a is responsible for driving the observed phenotype. As previously mentioned, the effect of PGC1a appears to be primarily derived from tumor extrinsic factors, specifically those originating from hepatic cells. However, we have not yet conducted further investigations to determine whether this effect is mediated through the cytosolic or nuclear pool of PGC1a. We will conduct additional studies in the near future to determine the specific form of PGC1a that is involved in driving the observed phenotype.

Clear.

Comments: 6. Including information about PA and PGC1a concentrations under normal physiological and pathological conditions (such as NAFLD) would enhance the context.

Response: Thank you for your comment. PA has been shown to have hepatotoxic effects, contributing to the development of hepatic steatosis and inflammation. Additionally, our studies have demonstrated that PA can enhance the expression and activity of PGC1a in liver cells. PGC1a is a key regulator of mitochondrial biogenesis and function, as well as other metabolic processes. In our study, we aimed to investigate the specific mechanisms through which PA enhances PGC1a in the context of lipid accumulation and hepatotoxicity. Furthermore, it is worth noting that PGC1a has been implicated in tumorigenic effects that promote breast cancer development. By elucidating these mechanisms, we hope to gain a better understanding of the role of PA and PGC1a in the pathogenesis of liver diseases such as NAFLD as well as breast cancer. Ultimately, this knowledge may contribute to the development of targeted therapies for hepatotoxicity and breast cancer. It has been included in the discussion.

What’s the concentration of PA under normal physiological and pathological conditions (such as NAFLD) would enhance the context?

 Minor:

Comments: 1. Ensure that all images have scale bars to aid readers in understanding the size of the depicted objects.

Response: Thank you for your comment. We appreciate your suggestion to include scale bars in all images or caption to aid readers in understanding the size of the depicted objects. We have included them.

Maybe the scale bar in figure 1G.

Comments: 2. Correct the legend of Figure 4C/D to reflect "PA-treated CM" instead of "PA (0.8mM)" to avoid misleading readers.

Response: Thank you for your suggestion. We have made changes to the legend based on the comments.

Comments: 3. Addressing these concerns and making these improvements will enhance the quality and clarity of the study before publication.

Response: Thank you for your valuable feedback. We hope that your major and minor suggestions will enhance the clarity and interpretation of our data, and ultimately improve the overall quality of our manuscript. We appreciate you bringing these issues to our attention, and we will work diligently to address each of them. Thank you again for your time and effort in reviewing our manuscript.

Minor editing of English language required
